# The Parameterized Complexity of Computing the VC-Dimension

**Florent Foucaud**
Université Clermont Auvergne,
CNRS, Mines Saint-Étienne,
Clermont Auvergne INP, LIMOS
Clermont-Ferrand, France
`florent.foucaud@uca.fr`

**Harmender Gahlawat**
Université Clermont Auvergne,
CNRS, Mines Saint-Étienne,
Clermont Auvergne INP, LIMOS
Clermont-Ferrand, France
`harmendergahlawat@gmail.com`

**Fionn Mc Inerney**
Telefónica Scientific Research
Barcelona, Spain
`fmcinern@gmail.com`

**Prafullkumar Tale**
Indian Institute of Science Education and Research Pune
Pune, India
`prafullkumar@iiserpune.ac.in`

## Abstract

The VC-dimension is a well-studied and fundamental complexity measure of a set system (or hypergraph) that is central to many areas of machine learning. We establish several new results on the complexity of computing the VC-dimension. In particular, given a hypergraph $\mathcal{H} = (\mathcal{V}, \mathcal{E})$, we prove that the naive $2^{\mathcal{O}(|\mathcal{V}|)}$-time algorithm is asymptotically tight under the Exponential Time Hypothesis (ETH). We then prove that the problem admits a 1-additive fixed-parameter approximation algorithm when parameterized by the maximum degree of $\mathcal{H}$ and a fixed-parameter algorithm when parameterized by its dimension, and that these are essentially the only such exploitable structural parameters. Lastly, we consider a generalization of the problem, formulated using graphs, which captures the VC-dimension of both set systems and graphs. We design a $2^{\mathcal{O}(\text{tw} \cdot \log \text{tw})} \cdot |V|$-time algorithm for any graph $G = (V, E)$ of treewidth $\text{tw}$ (which, for a set system, applies to the treewidth of its incidence graph). This is in contrast with closely related problems that require a double-exponential dependency on the treewidth (assuming the ETH).

## 1 Introduction

Vapnik and Chervonenkis introduced the *Vapnik-Chervonenkis Dimension* (VC-dimension) [75] as a measure of the richness of the expressivity of a set system. Given a non-empty finite set $\mathcal{V}$ and a set system $\mathcal{C} \subseteq 2^{\mathcal{V}}$, the *VC-dimension* of $\mathcal{C}$ is the size of a largest subset $S \subseteq \mathcal{V}$ that is *shattered* by $\mathcal{C}$, i.e., such that $\{C \cap S : C \in \mathcal{C}\} = 2^S$. The VC-dimension has proven to be immensely important in numerous fields, including machine learning, geometry, and combinatorics. Notably, the VC-dimension is intrinsic to several prominent topics in machine learning, such as $\epsilon$-nets [47], sample compression schemes [56], and machine teaching [44, 45].

In particular, $\epsilon$-nets are well-studied in learning theory (see, e.g., [2, 6, 7, 46, 66]) and have applications in the adversarial robustness of machine learning models (see, e.g., [24, 59]), with the seminal paper by Haussler and Welzl [47] proving the famous $\epsilon$-*net theorem*, which roughly states that set systems with fixed VC-dimension admit small $\epsilon$-nets. Furthermore, in relation to Valiant's probably approximately correct (PAC) learning [74], the VC-dimension is at the heart of one of the oldest open problems in machine learning: the *sample compression conjecture* of Floyd and Warmuth [37].

39th Conference on Neural Information Processing Systems (NeurIPS 2025).

This problem asks whether every set system of VC-dimension $d$ admits a sample compression scheme of size $\mathcal{O}(d)$, and it is actively pursued to this day, with a plethora of works making progress on it over the years (see, e.g., [4, 5, 16, 17, 18, 37, 48, 61, 62, 67]). The VC-dimension is also central to various machine teaching models and their major open problems. Specifically, Simon and Zilles asked whether every set system of VC-dimension $d$ has recursive teaching dimension $\mathcal{O}(d)$ [73], and Kirkpatrick, Simon, and Zilles asked whether every such set system has non-clashing teaching dimension at most $d$ [52], with numerous works (see, e.g., [15, 19, 26, 36, 42, 49, 57, 60, 71, 72]) making advances toward these questions and related directions. Overall, the fundamental nature of the VC-dimension in these various areas has motivated the study of its computational complexity, with the associated decision problem often equivalently formulated using hypergraphs as follows.

---

VC-DIMENSION
**Input**: A hypergraph $\mathcal{H} = (\mathcal{V}, \mathcal{E})$ and a positive integer $k$.
**Question**: Does there exist a subset $S \subseteq \mathcal{V}$ such that $|S| \geq k$ and $\{S \cap e : e \in \mathcal{E}\} = 2^S$?

---

It is easy to see that VC-DIMENSION can be solved in $|\mathcal{H}|^{\mathcal{O}(\log |\mathcal{H}|)}$ time, i.e., quasi-polynomial time, and thus, is most likely not NP-hard. Papadimitriou and Yannakakis [68] introduced the complexity class LogNP and proved that VC-DIMENSION is LogNP-complete. As LogNP lies in between P and NP, and both inclusions are believed to be proper, it is unlikely that VC-DIMENSION is in P either. Their result also implies that, assuming the Exponential Time Hypothesis (ETH) (the ETH is a standard conjecture in computational complexity), VC-DIMENSION cannot be solved in $|\mathcal{H}|^{o(\log |\mathcal{H}|)}$ time. Recently, Manurangsi [58] proved that VC-DIMENSION is highly inapproximable under well-established complexity-theoretic hypotheses, even ruling out a polynomial-time approximation algorithm with $o(\log |\mathcal{H}|)$ approximation factor, assuming the Gap-ETH (a strengthening of the ETH).

**Our Contributions.** We make further progress on the complexity of computing the VC-dimension. We first complement the result of Papadimitriou and Yannakakis by proving that the naive brute-force $2^{\mathcal{O}(|\mathcal{V}|)}$-time algorithm[1] for VC-DIMENSION— that tests each possible subset of $\mathcal{V}$ to see whether it is shattered — has a tight running time under the ETH (Theorem 9).

Together with the hardness results from the literature, this motivates studying the *parameterized complexity* [25, 29] of VC-DIMENSION. Indeed, this paradigm allows for a refined analysis of the computational complexity of a problem by measuring its complexity not only with respect to the input size $I$, but also an integer parameter $\ell$ that either originates from the problem formulation or captures well-defined structural properties of the input. The aim for such a parameterized problem is to design an algorithm solving it in $f(\ell) \cdot |I|^{\mathcal{O}(1)}$ time for a computable function $f$; this is known as a *fixed-parameter algorithm*. Parameterized problems that admit such an algorithm are called *fixed-parameter tractable* (FPT) with respect to the considered parameter. Under standard complexity assumptions, parameterized problems that are hard for the complexity class W[1] are not FPT.

In the above setting, Downey, Evans, and Fellows [28] proved that VC-DIMENSION is W[1]-complete when parameterized by the solution size $k$, and Manurangsi [58] even ruled out an FPT approximation algorithm with factor $o(k)$, assuming the Gap-ETH. However, apart from a result of Drange, Greaves, Muzi, and Reidl [30] proving that VC-DIMENSION is W[1]-hard parameterized by the degeneracy of the hypergraph $\mathcal{H}$, little is known about structural parameterizations of VC-DIMENSION.

We perform a systematic analysis in this direction. We prove that there is an FPT 1-additive approximation algorithm for VC-DIMENSION parameterized by the maximum degree $\Delta$ of $\mathcal{H}$ (Theorem 12), i.e., it computes a shattered set of size at least VC-dimension minus one. It can also be observed that VC-DIMENSION is FPT parameterized by the dimension $D$ of $\mathcal{H}$, i.e., the maximum size of a hyperedge in $\mathcal{H}$. Indeed, any shattered set is contained within a hyperedge of $\mathcal{H}$. Thus, one can test whether any of the at most $2^D$ possible subsets of any hyperedge is shattered in $2^D \cdot |\mathcal{H}|^{\mathcal{O}(1)}$ time. Unfortunately, we prove that the remaining core structural hypergraph parameters (hypertreewidth and transversal number) do not yield FPT algorithms for VC-DIMENSION (Proposition 13).

In search of more exploitable structural parameters, we turn our attention to the VC-dimension in graphs. The VC-dimension of a graph $G = (V, E)$ is the VC-dimension of the set system whose ground set is $V$ and whose sets are all the open neighborhoods of the vertices in $V$. The problem GRAPH-VC-DIMENSION is defined similarly to VC-DIMENSION, but for graphs: it takes a graph $G$ and an integer $k$ as input, and asks whether the VC-dimension of $G$ is at least $k$.

---

[1] Note that $|\mathcal{H}| = 2^{\mathcal{O}(|\mathcal{V}|)}$ and testing whether a set is shattered can be done in polynomial time.

GRAPH-VC-DIMENSION is relevant for the numerous applications in machine learning where the data is inherently graph-structured, see, e.g., [63]. GRAPH-VC-DIMENSION is also interesting since instances of VC-DIMENSION can be converted to instances of GRAPH-VC-DIMENSION: indeed, any set system can be equivalently represented by a set of open neighborhoods in a bipartite graph (see, e.g., [52]) or a set of closed neighborhoods in a split graph (see, e.g., [16]). Given a set system $\mathcal{C} \subseteq 2^{\mathcal{V}}$, the graph $G$ can be created as follows: for all $C \in \mathcal{C}$, there is a vertex $x_C$, and, for all $v \in \mathcal{V}$, there is a vertex $v$, with $x_C$ adjacent to $v$ if and only if $v \in C$. Then, $\mathcal{C}$ is equivalent to the set of open neighborhoods of vertices in $Y := \{x_C : C \in \mathcal{C}\}$ in the bipartite graph $G$. For closed neighborhoods, one needs to make the vertices in $Y$ form a clique (making $G$ a split graph) and take the closed neighborhoods of the vertices in $Y$. Both of these set systems are well-established: for open neighborhoods, see, e.g., [9, 40, 41, 51, 52, 64], and for closed neighborhoods, see, e.g., [1, 3, 10, 16, 22, 31, 32, 47, 55]. The VC-dimension of other graph-related set systems has also been considered in [11, 15, 16, 20, 21, 33, 34, 55, 72]. See [22, 30] for experimental evaluations. Due to the equivalence above, we focus on the VC-dimension of open neighborhoods,[2] and study a generalization of both VC-DIMENSION and GRAPH-VC-DIMENSION formulated using graphs:

---

GENERALIZED VC-DIMENSION (GEN-VC-DIMENSION)
**Input**: A graph $G = (V, E)$, two subsets $X, Y \subseteq V$, and a positive integer $k$.
**Question**: Does there exist a subset $S \subseteq X$ such that $|S| \geq k$ and $\{S \cap N(y) : y \in Y\} = 2^S$?

---

Indeed, GRAPH-VC-DIMENSION and VC-DIMENSION are special cases of GEN-VC-DIMENSION: any instance $(G, k)$ of GRAPH-VC-DIMENSION corresponds to an instance $(G, X = V, Y = V, k)$ of GEN-VC-DIMENSION, and VC-DIMENSION considers the bipartite incidence graph $G$ of the input set system (as defined above), with $X = \mathcal{V}$ and $Y = \{x_C : C \in \mathcal{C}\}$. Hence, our algorithms for GEN-VC-DIMENSION apply to both VC-DIMENSION and GRAPH-VC-DIMENSION.

We focus on the treewidth parameterization, which has been exploited for various machine learning problems, see, e.g., [12, 27, 35, 43, 65, 69]. Treewidth is arguably the most successful graph parameter due to the celebrated Courcelle's theorem [23], which states that any graph property definable in monadic second-order logic can be checked in linear time for graphs of bounded treewidth. In fact, this is the case for GEN-VC-DIMENSION [38, Theorem 6.5]. However, the obtained dependency on the treewidth is a tower of exponentials whose height is a function of the treewidth.

We design a much faster $2^{\mathcal{O}(\text{tw} \cdot \log \text{tw})} \cdot |V|$-time algorithm for GEN-VC-DIMENSION (Theorem 19), where tw is the treewidth of $G$. In particular, this result applies both to GRAPH-VC-DIMENSION and to VC-DIMENSION (where the considered graph is the bipartite incidence graph representation of the set system as described earlier). Thus, this further motivates considering the VC-dimension of open neighborhoods in graphs rather than closed neighborhoods. Indeed, as the treewidth of a split graph is one less than the size of its largest clique, then when considering closed neighborhoods in a split graph $G$, we have that $G$ has small treewidth if and only if $Y$ is small. In contrast, for open neighborhoods in a bipartite graph $G$, it can be that $Y$ is very large even if $G$ has small treewidth. We also emphasize that the running time of our fixed-parameter algorithm has a relatively low dependency on the treewidth that contrasts with (tight) double-exponential dependencies on the treewidth experienced by recently studied and closely related problems [14, 39].

Finally, this FPT algorithm is complemented by a lower bound ruling out an algorithm for GRAPH-VC-DIMENSION running in $2^{o(\text{vcn}+k)} \cdot |V|^{\mathcal{O}(1)}$ time (Theorem 9), where $k$ is the solution size and vcn is the vertex cover number of $G$, an even larger parameter than the treewidth of $G$ as vcn $\geq$ tw.

## 2 Preliminaries

For $\ell \in \mathbb{N}$, let $[\ell] = \{1, \dots, \ell\}$ and $[0, \ell] = \{0\} \cup [\ell]$. Let $\mathbb{N}_0 = \mathbb{N} \cup \{0\}$.

**Hypergraphs and Graphs.** Let $\mathcal{H} = (\mathcal{V}, \mathcal{E})$ be a hypergraph. Set $|\mathcal{H}| := |\mathcal{V}| + |\mathcal{E}|$. For a vertex $v \in \mathcal{V}$, $\text{inc}(v)$ is the set of edges that contain (or are *incident*) to $v$. The *degree* of $v$ in $\mathcal{H}$ is $\deg_{\mathcal{H}}(v) := |\text{inc}(v)|$. The maximum degree of $\mathcal{H}$ is $\Delta(\mathcal{H}) := \max_{v \in \mathcal{V}} \deg_{\mathcal{H}}(v)$ (or simply $\Delta$). The *transversal number* of $\mathcal{H}$ is the minimum size of a subset $X \subseteq \mathcal{V}$ such that $X \cap e \neq \emptyset$ for all $e \in \mathcal{E}$. If there exists a tree $T$ such that, for all $e \in \mathcal{E}$, $e$ is the set of vertices of a subtree of $T$, then $\mathcal{H}$ is a *hypertree* [13]. Let $G = (V, E)$ be a graph. The *open neighborhood* of a vertex $v \in V$ is the set

---

[2]We remark that, with minor modifications, many of our results also hold for closed neighborhoods.

$N(v) := \{u \mid uv \in E\}$, and the degree of $v$ in $G$ is $d_G(v) := |N(v)|$. For a subset $X \subseteq V$, let $N_X(v) = N(v) \cap X$. The *closed neighborhood* of a vertex $v \in V$ is the set $N[v] := N(v) \cup \{v\}$. For a subset $X \subseteq V$, let $G[X]$ denote the subgraph of $G$ induced by the vertices in $X$, and let $G - X$ denote the subgraph $G[V \setminus X]$ of $G$. A subset $U \subseteq V$ is a *vertex cover* of $G$ if, for each edge in $G$, at least one of its endpoints is in $U$. The minimum cardinality of a vertex cover of $G$ is its *vertex cover number* (vcn). A subset $S \subseteq V$ is a *separator* if $G - S$ contains at least two connected components.

A *tree-decomposition* of a graph $G$ is a pair $(T, \beta)$, where $T$ is a tree and $\beta : V(T) \to 2^{V(G)}$ is a mapping from $V(T)$ (called *bags*) to subsets of $V(G)$ such that:

1. for all $uv \in E(G)$, there exists $t \in V(T)$ such that $\{u, v\} \subseteq \beta(t)$;

2. for all $v \in V(G)$, the subgraph of $T$ induced by $T_v = \{t \in V(T) \mid v \in \beta(t)\}$ is a non-empty tree.

The *width* of a tree-decomposition $(T, \beta)$ is $\max_{t \in V(T)} |\beta(t)| - 1$. The *treewidth* of $G$, denoted by $\mathrm{tw}(G)$ (or simply tw), is the minimum possible width of a tree-decomposition of $G$. The mapping $\beta$ can be extended from vertices of $T$ to subgraphs of $T$: for a subgraph $U$ of $T$, $\beta(U) = \bigcup_{v \in V(U)} \beta(v)$. Computing a tree-decomposition of minimum width is FPT parameterized by the treewidth [8, 53], and a tree-decomposition of width at most 2tw can be computed in $2^{\mathcal{O}(\mathrm{tw})} \cdot |V(G)|$ time [54]. For dynamic programming, it is useful to have a tree-decomposition with additional properties. A tree-decomposition $(T, \beta)$ is *nice* if each node $t \in V(T)$ is exactly one of the following four types:

1. **Leaf:** $t$ is a leaf of $T$ and $|\beta(t)| = 0$.

2. **Introduce:** $t$ has a unique child $c$ and there exists $v \in V(G)$ such that $\beta(t) = \beta(c) \cup \{v\}$.

3. **Forget:** $t$ has a unique child $c$ and there exists $v \in V(G)$ such that $\beta(c) = \beta(t) \cup \{v\}$.

4. **Join:** $t$ has exactly two children $c_1, c_2$ and $\beta(t) = \beta(c_1) = \beta(c_2)$.

Let $(T, \beta)$ be a (nice) tree-decomposition of $G$ and $t$ a node of $T$. The subtree of $T$ rooted at $t$ is $T_t$, $G_t = G[T_t]$, $G_t^{\uparrow} = G - V(G_t)$, and $G_t^{\downarrow} = G_t - \beta(t)$. Given a tree-decomposition, we can obtain a nice tree-decomposition of the same width and with a linear number of nodes in linear time [25]. Thus, a nice tree-decomposition of a graph $G$ of width at most 2tw and with $\mathcal{O}(V(G))$ nodes can be computed in $2^{\mathcal{O}(\mathrm{tw})} \cdot |V(G)|$ time. So, if we seek an algorithm with at least that runtime, we may assume that such a nice tree-decomposition is part of the input.

**VC-Dimension.** Let $\mathcal{H} = (\mathcal{V}, \mathcal{E})$ be a hypergraph. A set $U \subseteq \mathcal{V}$ is a *shattered set* if, for all $U' \subseteq U$, there exists $e \in \mathcal{E}$ such that $U \cap e = U'$. The *VC-dimension of* $\mathcal{H}$ is the maximum size of a shattered set of $\mathcal{H}$. For $A \subseteq \mathcal{V}$ and $A' \subseteq A$, an edge $e$ *witnesses* $A'$ in $A$ if $e \cap A = A'$. Let $U$ be a shattered set of $\mathcal{H}$. Then, there exists a set $\mathcal{E}' \subseteq \mathcal{E}$ such that, for all $U' \subseteq U$, there exists an edge $e \in \mathcal{E}'$ with $e$ witnessing $U'$ in $U$. We call such an $\mathcal{E}'$ a *shattering set* of $U$, and a minimal shattering set $W \subseteq \mathcal{E}$ of $U$ is said to be a *witness* of $U$. Observe that $|W| = 2^{|U|}$. In the context of the VC-dimension of a set $\mathcal{N}$ of open neighborhoods in a graph $G = (V, E)$, a subset of vertices $U \subseteq V$ is a *shattered set* if, for each subset $U' \subseteq U$, there exists a vertex $w$ that witnesses $U'$, i.e., $N(w) \in \mathcal{N}$ and $N_U(w) = U'$.

## 3 Algorithmic Lower Bounds

In this section, via a reduction from 3-COLORING to GRAPH-VC-DIMENSION, we establish that, assuming the ETH[3] [50], GRAPH-VC-DIMENSION cannot be solved in $2^{o(\mathrm{vcn}+k)} \cdot |V|^{\mathcal{O}(1)}$ time, where vcn is the vertex cover number of $G$. This same reduction proves that, assuming the ETH, VC-DIMENSION cannot be solved in $2^{o(|\mathcal{V}|)} \cdot |\mathcal{H}|^{\mathcal{O}(1)}$ time. An instance of 3-COLORING consists of a graph $G'$, and asks whether $G'$ admits a proper coloring with 3 colors. The following is well-known:

**Proposition 1** ([25]). *Assuming the* ETH*, there exists a constant $\epsilon_c > 0$ such that* 3-COLORING *does not admit an algorithm running in $2^{\epsilon_c \cdot |V(G')|} \cdot |V(G')|^{\mathcal{O}(1)}$ time.*

**Reduction.** Given an instance $G'$ of 3-COLORING, in time exponential in $|V(G')|$ (i.e., $2^{\mathcal{O}(|V(G')|)}$ time, but the value of the constant will be refined later), our reduction returns an instance $(G, k)$ of GRAPH-VC-DIMENSION. We note that it is expected that this reduction takes super-polynomial time since 3-COLORING is NP-hard, while VC-DIMENSION can be solved in quasi-polynomial

---

[3]The ETH roughly states that $n$-variable and $m$-clause 3-SAT cannot be solved in $2^{o(n+m)}$ time.

time, and hence, a polynomial-time reduction would imply that every problem in NP can be solved in quasi-polynomial time. Now, for the reduction, fix a constant $0 < \epsilon_1 < 1$, set $k = \lceil \epsilon_1 |V(G')| \rceil$, and arbitrarily partition the vertices of $V(G')$ into $k$ parts $V_1, \ldots, V_k$, each of size at most $p = \lceil \frac{1}{\epsilon_1} \rceil$. Note that, for all $i \in [k]$, there are at most $3^p$ possible 3-colorings of $G'[V_i]$. For each $V_i$, enumerate each valid coloring of $G'[V_i]$, and add a vertex corresponding to it in $U_i$ (which is in $G$). Thus, in $G$, for all $i \in [k]$, we have an independent set of vertices $U_i$ such that $|U_i| \leq 3^p$. Set $X := \bigcup_{i \in [k]} U_i$. We ensure that $X$ is a vertex cover of $G$. Thus, $G$ may be partitioned into $X$ and an independent set $Y$. We now specify the vertices in $Y$, which we partition into three sets $I_1, I_2$, and $I_{\geq 3}$. For each vertex $u \in X$, we add a vertex $w \in I_1$ adjacent to $u$. For all distinct $i, j \in [k]$, $u \in U_i$, and $v \in U_j$, we add a vertex $w \in I_2$ and make it adjacent to $u$ and $v$ if and only if $u$ and $v$ are *consistent with each other*, by which we mean the union of their corresponding colorings is a proper coloring of $G'[V_i \cup V_j]$. For each $A \subseteq [k]$ with $|A| \geq 3$, we add a vertex $w \in I_{\geq 3}$ adjacent to each vertex in $U_j$ for $j \in A$. This completes our reduction (see Figure 1). We now prove its correctness.

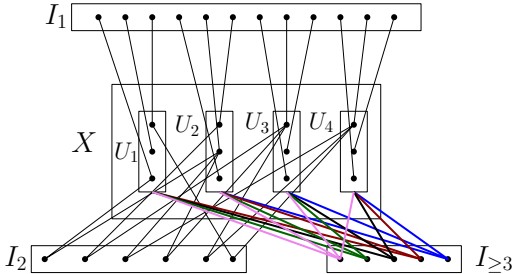

Figure 1: The graph $G$ constructed from $G'$, where $X$ (vertex cover of $G$) and $Y = I_1 \cup I_2 \cup I_{\geq 3}$ are independent sets. An edge between $w \in I_{\geq 3}$ and $U_i$ signifies that each vertex in $U_i$ is adjacent to $w$.

**Lemma 2.** *If $G'$ admits a proper 3-coloring, then $G$ contains a shattered set of size $k$.*

*Proof.* If $G'$ admits a proper 3-coloring, then there exist $u_1, \ldots, u_k \in V(G)$ such that $u_i \in U_i$ for $i \in [k]$ and $u_1, \ldots, u_k$ are mutually consistent. We claim that $S = \{u_1, \ldots, u_k\}$ is a shattered set in $G$. First, by the definition of $I_1$, for each $u_i$, there is a unique vertex $w_i \in I_1$ such that $N(w_i) = u_i$. Second, by the definition of $I_2$ and the fact that the vertices in $S$ are mutually consistent, for every two distinct vertices $u_i, u_j$ in $S$, there is a unique vertex $w \in I_2$ such that $N(w) = \{u_i, u_j\}$. Third, by the definition of $I_{\geq 3}$ and the fact that each $u_i$ comes from a distinct $U_i$, for every subset $A \subseteq S$ such that $|A| \geq 3$, there is a unique vertex $w \in I_{\geq 3}$ such that $N_S(w) = A$. Finally, since we can assume that $|X| > k$ (as $k \leq |V(G')|$ and $X$ encodes all possible 3-colorings of $|V(G')|$ vertices), there is at least one vertex $w \in I_1$ such that $N_S(w) = \emptyset$. Thus, $S$ is a shattered set of size $k$ in $G$. $\square$

**Lemma 3.** *If $G$ contains a shattered set of size at least $k$, then $G'$ admits a proper 3-coloring.*

*Proof.* For clarity, we divide this proof into multiple claims. Let $S$ be a shattered set of $G$ such that $|S| \geq k$. Since $G$ is a bipartite graph with bipartition $X \cup Y$, either $S \subseteq X$ or $S \subseteq Y = I_1 \cup I_2 \cup I_{\geq 3}$.

**Claim 4.** $S \subseteq X$.

*Proof of Claim.* Toward a contradiction, let $S \subseteq Y$. For all $v \in S$, the degree of $v$ is at least $2^{k-1}$, and thus, $v \in I_{\geq 3}$ as the degree of each vertex in $I_1$ and $I_2$ is at most 2. Recall that if a vertex $v \in I_{\geq 3}$ is adjacent to some vertex in $U_i$, then it is adjacent to each vertex in $U_i$. Hence, any minimal set of witnesses for $S$ contains at most one vertex from each $U_i$. Thus, $|S| \leq \log k < k$, a contradiction. $\diamond$

**Claim 5.** *For all $i \in [k]$, $|S \cap U_i| < 3$.*

*Proof of Claim.* Toward a contradiction, suppose that there exists an $i \in [k]$ such that $|S \cap U_i| \geq 3$. Let $x, y, z \in S \cap U_i$ be distinct. As they are in $S$, for each subset $A$ of $\{x, y, z\}$, there is a vertex $w \in Y$ (as $X$ is an independent set) such that $N_S(w) = A$. Thus, there is a vertex $w \in Y$ such that $x, y \in N(w)$, but $z \notin N(w)$. As each vertex in $I_1$ has degree 1, and $w$ has degree at least 2, then $w \notin I_1$. As each vertex in $I_2$ cannot be adjacent to two vertices in the same $U_i$, then $w \notin I_2$. If a vertex in $I_{\geq 3}$ is adjacent to some vertex in $U_i$, then it is adjacent to each vertex in $U_i$, and hence, it is not possible that $w \in I_{\geq 3}$ either. Thus, we have a contradiction. $\diamond$

**Claim 6.** *For distinct $i, j \in [k]$, it is not possible that $|S \cap U_i| > 1$ and $|S \cap U_j| > 1$.*

*Proof of Claim.* Toward a contradiction, assume that there are distinct $i, j \in [k]$ such that $|S \cap U_i| > 1$ and $|S \cap U_j| > 1$. Let $a, b \in S \cap U_i$ and let $x, y \in S \cap U_j$. Then, there is $w \in Y$ such that $a, b, x \in N(w)$, but $y \notin N(w)$. As in the proof of Claim 5, $w \notin I_1 \cup I_2$ due to the degree arguments, and it is not possible that $x \in N(w)$ but $y \notin N(w)$ if $w \in I_{\geq 3}$. Thus, we have a contradiction. $\diamond$

By Claim 4, $S \subseteq X$, and by Claims 5 and 6, either $S$ contains exactly one vertex from each $U_i$ (for $i \in [k]$) or there is at most one $j \in [k]$ such that $|S \cap U_j| = 2$, and, for all $i \in [k]$ such that $i \neq j$, $|S \cap U_i| \leq 1$. Next, we show that the latter case is not possible.

**Claim 7.** *For all $i \in [k]$, $|S \cap U_i| = 1$.*

*Proof of Claim.* Toward a contradiction, suppose that there is a unique $j \in [k]$ such that $|S \cap U_j| = 2$, and, for all $i \in [k]$ such that $i \neq j$, $|S \cap U_i| \leq 1$. Note that there is at most one $U_\ell$, $\ell \in [k]$, such that $|S \cap U_\ell| = 0$. Let $x, y \in S \cap U_j$ be distinct. Then, there is a vertex $w \in Y$ such that $N_S(w) = \{x, y\}$. As in the proof of Claim 5, $w \notin I_1$ due to its degree, and $w \notin I_2$ as a vertex in $I_2$ cannot be adjacent to two vertices from the same $U_j$. Each vertex in $I_{\geq 3}$ is adjacent to each vertex of at least three sets in $\{U_1, \ldots, U_k\}$, and hence, to each vertex of at least two sets in $\{U_1, \ldots, U_k\} \setminus \{U_\ell\}$. Thus, if $x, y \in N_S(w)$, then there is at least one more vertex $v \in U_i \cap S$ where $i \in [k] \setminus \{j, \ell\}$ such that $v \in N_S(w)$. Hence, $N_S(w) \neq \{x, y\}$, as $v$ is distinct from $x$ and $y$. Thus, we have a contradiction. $\diamond$

By Claim 7, $S$ contains exactly one vertex from each $U_i$. To complete our proof, we show that the vertices in $S$ are mutually consistent. Toward a contradiction, assume that there are distinct $x, y \in S$ such that $x$ and $y$ are not consistent. Let $x \in U_i$ and $y \in U_j$ for distinct $i, j \in [k]$. As $x, y \in S$, there exists $w \in Y$ such that $N_S(w) = \{x, y\}$. As each vertex in $I_1$ has degree 1, $w \notin I_1$. Since each vertex in $I_{\geq 3}$ is adjacent to each vertex of at least three sets in $\{U_1, \ldots, U_k\}$, and $S$ contains at least one vertex from each $U_i$, we have that $w \notin I_{\geq 3}$. So, $w \in I_2$. By the definition of $I_2$, if $w$ is adjacent to $x$ and $y$, then $x$ and $y$ are consistent. Thus, all vertices in $S$ are mutually consistent, and so, a coloring corresponding to these vertices is a proper 3-coloring of $G'$. $\square$

**Lemma 8.** *There exists a constant $\epsilon > 0$ such that, if the instance $(G, k)$ of* GRAPH-VC-DIMENSION *is solvable in $2^{\epsilon \cdot \mathsf{vcn}} \cdot |V(G)|^{\mathcal{O}(1)}$ time, then the instance $G'$ of* 3-COLORING *is solvable in $2^{\epsilon_c \cdot |V(G')|} \cdot |V(G')|^{\mathcal{O}(1)}$ time.*

*Proof.* Note that $|X| \leq 3^p \cdot k = 3^{\lceil \frac{1}{\epsilon_1} \rceil} \lceil \epsilon_1 \cdot |V(G')| \rceil$, $|I_1| = |X|$, $|I_2| \leq |X|^2$, and $|I_{\geq 3}| = \binom{k}{\geq 3} \leq 2^k$. Thus, $|V(G)| = |X| + |I_1| + |I_2| + |I_{\geq 3}| \leq 2|X| + |X|^2 + 2^k \leq 3|X|^2 + 2^k$. Further, we can assume that $3(3^{\lceil \frac{1}{\epsilon_1} \rceil} \cdot \lceil \epsilon_1 \cdot |V(G')| \rceil)^2 \leq 2^{\lceil \epsilon_1 \cdot |V(G')| \rceil}$, as otherwise $|V(G')|$ is bounded by a constant and we can solve the instance of 3-COLORING by brute force. So, $|V(G)| \leq 2 \cdot 2^{\lceil \epsilon_1 \cdot |V(G')| \rceil} \leq 2^{\epsilon_1 |V(G')| + 2}$.

First, we construct $(G, k)$ from $G'$ in $|V(G)|^{\mathcal{O}(1)}$ time. As the instances $(G, k)$ of GEN-VC-DIMENSION and $G'$ of 3-COLORING are equivalent (by Lemmas 2 and 3), then solving $(G, k)$ solves $G'$ too. Assume that we can solve $(G, k)$ in $2^{\epsilon \cdot \mathsf{vcn}} |V(G)|^{\mathcal{O}(1)}$ time, and thus, can solve 3-COLORING on $G'$ in $2^{\epsilon \cdot \mathsf{vcn}} |V(G)|^{\mathcal{O}(1)} + |V(G)|^{\mathcal{O}(1)}$ time. There exists a constant $c > 1$ such that $2^{\epsilon \cdot \mathsf{vcn}} |V(G)|^{\mathcal{O}(1)} + |V(G)|^{\mathcal{O}(1)} \leq 2^{\epsilon \cdot \mathsf{vcn}} |V(G)|^c$. Recall that $X$ is a vertex cover of $G$. Thus, our running time is at most $2^{\epsilon \cdot 3^{\lceil \frac{1}{\epsilon_1} \rceil} \lceil \epsilon_1 \cdot |V(G')| \rceil} \cdot 2^{c(\epsilon_1 |V(G')| + 2)}$. Fix $\epsilon = \frac{1}{3^{\lceil \frac{1}{\epsilon_1} \rceil}}$. We can safely assume that $\lceil \epsilon_1 |V(G')| \rceil \leq 2\epsilon_1 |V(G')|$ and $\epsilon_1 |V(G')| + 2 \leq 2\epsilon_1 |V(G')|$. Replacing these values (including the value of $\epsilon$) in our running time, it is at most $2^{2\epsilon_1 \cdot |V(G')|} \cdot 2^{2c\epsilon_1 \cdot |V(G')|} = 2^{|V(G')|(2\epsilon_1 + 2c\epsilon_1)}$. Since $c > 1$, our running time is at most $2^{|V(G')|(4c\epsilon_1)}$. For any such constant $c$, we can set $0 < \epsilon_1 < 1$ such that $4c\epsilon_1 < \epsilon_c$. Thus, if we can solve $(G, k)$ in $2^{\epsilon \mathsf{vcn}} \cdot |V(G)|^{\mathcal{O}(1)}$ time, then we can solve 3-COLORING on $G'$ in $2^{\epsilon_c \cdot |V(G')|} \cdot |V(G')|^{\mathcal{O}(1)}$ time. $\square$

Finally, the following theorem is a consequence of our reduction, Lemmas 2, 3, and 8, and Proposition 1. Also, since $k \leq \mathsf{vcn}$ in our construction, the result for GRAPH-VC-DIMENSION holds for the combined parameter $k + \mathsf{vcn}$. Moreover, the lower bound for VC-DIMENSION follows since, from our instance $(G, k)$ of GRAPH-VC-DIMENSION (recall that $G$ is a bipartite graph with bipartition

$X \cup Y$), we can create an equivalent instance $(\mathcal{H} = (\mathcal{V}, \mathcal{E}), k)$ of VC-DIMENSION in polynomial time as follows: set $\mathcal{V} = X$ and, for all $y \in Y$, create a hyperedge containing $N(Y)$.

**Theorem 9.** *Assuming the* ETH*, there exists a constant $\epsilon > 0$ such that the following statements hold: (i)* GRAPH-VC-DIMENSION *does not admit an algorithm running in $2^{\epsilon(\mathsf{vcn}+k)}(|V|)^{\mathcal{O}(1)}$ time, and (ii)* VC-DIMENSION *does not admit an algorithm running in $2^{\epsilon|\mathcal{V}|} \cdot |\mathcal{H}|^{\mathcal{O}(1)}$ time.*

## 4 Parameterized Complexity of VC-DIMENSION

In this section, we first provide an FPT 1-additive approximation algorithm for VC-DIMENSION parameterized by the maximum degree $\Delta := \Delta(\mathcal{H})$ of $\mathcal{H}$. As noted before, VC-DIMENSION is FPT parameterized by the dimension of $\mathcal{H}$ and W[1]-hard parameterized by the solution size $k$ [28] and the degeneracy of $\mathcal{H}$ [30]. We then cover the remaining well-studied structural hypergraph parameters by proving that VC-DIMENSION is LogNP-hard, even if $\mathcal{H}$ is a hypertree with transversal number 1.

Toward the approximation algorithm, we observe a relationship between a shattered set and its witness in $\mathcal{H} = (\mathcal{V}, \mathcal{E})$. Let $S := \{v_1, \ldots, v_k\} \subseteq \mathcal{V}$ be a shattered set of $\mathcal{H}$ of size $k$, and $W \subseteq \mathcal{E}$ a witness of $S$. Note that $|W| = 2^k$ and, for each subset $S' \subseteq S$, there exists a unique edge $e \in W$ such that $e \cap S = S'$. In other words, for each $A \subseteq [k]$, there exists an edge $e \in W$ such that $v_i \in e$ if and only if $i \in A$. Thus, there exists an ordering $(w_0, \ldots, w_{2^k-1})$ of the edges of $W$ such that $v_i \in w_j$ (for $i \in [k]$ and $j \in [0, 2^k - 1]$) if and only if the $i^{\text{th}}$ least significant bit of the binary representation of $j$ is 1. We call such an ordering a *good ordering*. We observe a useful property of this relationship.

**Lemma 10.** *Given a hypergraph $\mathcal{H} = (\mathcal{V}, \mathcal{E})$ and $W \subseteq \mathcal{E}$ with $|W| = 2^k$, one can decide whether there exists a shattered set $S \subseteq \mathcal{V}$ such that $|S| = k$ and $W$ is a witness of $S$ in $|W|^{|W|} \cdot |\mathcal{H}|^{\mathcal{O}(1)}$ time.*

*Proof.* The algorithm is the following. Enumerate all possible orderings of $W$ and, for each ordering $(w_0, \ldots, w_{2^k-1})$ of $W$, decide whether it is a *good ordering* for some subset $S \subseteq \mathcal{V}$ as follows: if, for each $i \in [k]$, there exists at least one vertex $v \in \mathcal{V}$ such that $w_j \in \mathsf{inc}(v) \cap W$ if and only if the $i^{\text{th}}$ least significant bit of $j$ is 1, then $W$ is a good ordering. From a good ordering, one can easily extract a shattered set $S$ which contains exactly one vertex satisfying the condition above for each $i \in [k]$. Since all possible orderings of $W$ can be enumerated in $\mathcal{O}(|W|^{|W|})$ time, and, for each ordering, it can be checked in polynomial time whether it is a good ordering, then it can be decided whether $W$ witnesses some shattered set of size $k$ in $|W|^{|W|} \cdot |\mathcal{H}|^{\mathcal{O}(1)}$ time. $\qquad\square$

**Observation 11.** *Let $S \subseteq \mathcal{V}$ be a non-trivial shattered set of a hypergraph $\mathcal{H} = (\mathcal{V}, \mathcal{E})$. Then, for each vertex $v \in S$, there exists a subset of $\mathsf{inc}(v)$ that shatters $S \setminus \{v\}$.*

*Proof.* Let $W \subseteq \mathcal{E}$ be a shattering set of $S$, and let $S_v = S \setminus \{v\}$. Then, for each subset $S' \subseteq S_v$, there is an edge $e \in W$ such that $S' \cup \{v\} = e \cap S$. Thus, for each subset $S' \subseteq S_v$, there exists an edge $e \in \mathsf{inc}(v)$ such that $e \cap S_v = S'$, and hence, $\mathsf{inc}(v)$ contains a subset that shatters $S_v$. $\qquad\square$

We now provide an FPT 1-additive approximation algorithm for VC-DIMENSION parameterized by $\Delta$. It applies the next algorithm at most $\log \Delta$ times, since the VC-dimension of $\mathcal{H}$ is at most $\log \Delta + 1$, as any vertex in a shattered set of size greater than $\log \Delta + 1$ would need to be in more than $2^{\log \Delta + 1 - 1} = \Delta$ hyperedges.

**Theorem 12.** VC-DIMENSION *admits a $2^{\mathcal{O}(\Delta \log \Delta)} \cdot |\mathcal{H}|^{\mathcal{O}(1)}$-time algorithm that either outputs a shattered set of size $k - 1$ or certifies that there is no shattered set of size $k$ in $\mathcal{H}$.*

*Proof.* If $\mathcal{H} = (\mathcal{V}, \mathcal{E})$ contains a shattered set $S$ of size $k$, then by Observation 11, there exists $v \in S \subseteq \mathcal{V}$ such that a subset of $\mathsf{inc}(v)$ shatters a set of size $k - 1$. For all $v \in \mathcal{V}$, we look at each subset $W \subseteq \mathsf{inc}(v)$ of size $2^{k-1}$ to decide whether $W$ witnesses a shattered set $S \subseteq \mathcal{V}$ of size $k - 1$ using Lemma 10, and report $S$ if it exists. The correctness of the algorithm follows from Observation 11 since, if there is a shattered set of size at least $k$, then our algorithm will report a shattered set of size $k - 1$, and if our algorithm fails to find a shattered set of size $k - 1$, then it would imply that there is no shattered set of size $k$ in $\mathcal{H}$. Finally, we analyze the running time. For all $v \in \mathcal{V}$, we look at all subsets of $\mathsf{inc}(v)$ of size $2^{k-1}$, and then, for each subset, we invoke the algorithm from Lemma 10. Since $|\mathsf{inc}(v)| \leq \Delta$ for each $v \in \mathcal{V}$, we consider at most $2^\Delta$ subsets of $\mathsf{inc}(v)$, and as

the size of each of these subsets is at most $\Delta$, for each of these subsets we need $\Delta^\Delta \cdot |\mathcal{H}|^{\mathcal{O}(1)}$ time. Thus, the total running time is $2^\Delta \cdot \Delta^\Delta \cdot |\mathcal{H}|^{\mathcal{O}(1)} = 2^{\mathcal{O}(\Delta \log \Delta)} \cdot |\mathcal{H}|^{\mathcal{O}(1)}$. $\qquad\square$

Theorem 12 also applies to GEN-VC-DIMENSION where the vertices in $X$ have maximum degree at most $\Delta$. When all vertices in $X \cup Y$ have maximum degree $\Delta$ (this applies for example to GRAPH-VC-DIMENSION for input graphs of maximum degree $\Delta$), then one can actually obtain an FPT algorithm running in $2^{\mathcal{O}(\Delta^2 \log \Delta)}|V|^{\mathcal{O}(1)}$ time [3, Theorem 15]. This can even be improved to $2^{\mathcal{O}(\log^2 \Delta)}|V|^{\mathcal{O}(1)}$: observe that the solution must be contained in the neighborhood of some vertex in $Y$; hence it suffices to enumerate, for each such neighborhood (which is of size at most $\Delta$), each of its possible subsets of size $\log \Delta + 1$ and check in polynomial time whether it is shattered.

However, the remaining well-known structural hypergraph parameters do not yield tractability:

**Proposition 13.** VC-DIMENSION *is LogNP-hard, even if $\mathcal{H}$ is a hypertree with transversal number 1.*

*Proof.* Given a hypergraph $\mathcal{H} = (\mathcal{V}, \mathcal{E})$, let $\mathcal{H}' = (\mathcal{V}', \mathcal{E}')$ be the hypergraph obtained from $\mathcal{H}$ by adding a vertex $u$ to each hyperedge. Then, $\mathcal{H}'$ has the same VC-dimension as $\mathcal{H}$, and $\mathcal{H}'$ is a hypertree with transversal number 1 (since $\{u\}$ is a transversal). To see that it is a hypertree, notice that the tree $T$ is the star centered at $u$ whose leaves are $\mathcal{V}$. Since for all $e \in \mathcal{E}'$, we have $u \in e$, the vertices of $e$ form a subtree of $T$. $\qquad\square$

## 5  Tractability via Treewidth

In this section, we provide a $2^{\mathcal{O}(\mathrm{tw} \cdot \log \mathrm{tw})} \cdot |V|$ algorithm to solve GEN-VC-DIMENSION, where $\mathrm{tw}$ denotes the treewidth of $G$. For the rest of this section, let $(G, X, Y, k)$ be an instance of GEN-VC-DIMENSION. First, we establish that if a shattered set intersects at least two components of $G - Z$ for some separator $Z$, then the shattered set is small compared to $|Z|$:

**Lemma 14.** *Let $S$ be a shattered set of $G$, $Z$ a separator of $G$, and $C_1, \ldots, C_p$ the components of $G - Z$. If there are distinct $i, j \in [p]$ such that $S \cap C_i \neq \emptyset$ and $S \cap C_j \neq \emptyset$, then $|S| \leq \log |Z| + 2$.*

*Proof.* Let $v \in V(C_i) \cap S$ and $u \in V(C_j) \cap S$. If a vertex $x$ witnesses a subset $A \subseteq S$ that contains both $u$ and $v$, then $x$ is in $Z$ (as $Z$ separates $C_i$ and $C_j$). As there are $2^{|S|-2}$ subsets of $S$ that contain both $u$ and $v$, there need to be at least $2^{|S|-2}$ witnesses in $Z$, i.e., $|S| \leq \log |Z| + 2$. $\qquad\square$

Let $(T, \beta)$ be a (nice) tree-decomposition of $G$ of width $\mathrm{tw}$, and let $S$ be a shattered set of $G$. Similarly to Lemma 14, either there exists a bag that completely contains $S$, or $S$ is small compared to $\mathrm{tw}$:

**Lemma 15.** *Let $(T, \beta)$ be a (nice) tree-decomposition of $G$ of width $\mathrm{tw}$, and let $S$ be a shattered set of $G$. Then, $|S| \leq \log \mathrm{tw} + 2$ or there exists some node $v \in V(T)$ such that $S \subseteq \beta(v)$.*

*Proof.* If there exists a bag that contains $S$, then we are done. Hence, we assume that there is no $v \in V(T)$ such that $S \subseteq \beta(v)$, and prove that $|S| \leq \log \mathrm{tw} + 2$. Without loss of generality, assume that $u$ and $w$ are two distinct vertices of $S$ that do not appear together in any bag of the tree-decomposition. Further, let $t_1$ and $t_2$ be two distinct nodes of $T$ such that $u \in \beta(t_1)$, $w \in \beta(t_2)$, and $t_1$ and $t_2$ are at minimum distance in $T$. Let $P$ be the unique $(t_1, t_2)$-path in $T$, and let $t$ be the neighbor of the node $t_1$ on $P$. Let $Z = \beta(t_1) \cap \beta(t)$. First, observe that, by our choice of $t, t_1$, and $t_2$, $u \notin \beta(t)$ and $w \notin \beta(t_1)$. Hence, $u, w \notin Z$. Moreover, observe that $Z$ is a separator in $G$ (due to the properties of tree-decompositions [25]) and $u$ and $w$ are vertices of $S$ that appear in distinct components of $G - Z$. Hence, by Lemma 14, we have that $|S| \leq \log |Z| + 2 \leq \log \mathrm{tw} + 2$. $\qquad\square$

We now proceed with the $2^{\mathcal{O}(\mathrm{tw} \cdot \log \mathrm{tw})} \cdot |V|$-time algorithm to solve GEN-VC-DIMENSION given a nice tree-decomposition $(T, \beta)$ of $G$. We compute a nice tree-decomposition of $G$ of width at most $2\mathrm{tw}$ and with $\mathcal{O}(|V|)$ nodes in time $2^{\mathcal{O}(\mathrm{tw})} \cdot |V|$ (see Section 2). Our algorithm has two phases.

**First Phase.** First, we assume that a shattered set of maximum size is contained in a bag of $T$. Using the technique of [30, Theorem 7],[4] after a pre-processing step that takes $2^{\mathcal{O}(\mathrm{tw})} \cdot |V|$ time, for each

---

[4]Note that this theorem considers the degeneracy of the graph for the running time, however, for any graph, its degeneracy is at most its treewidth. Thus, we obtain the runtimes we use here.

node $v \in V(T)$ and each subset $S \subseteq \beta(v) \cap X$, we check in $2^{\mathcal{O}(\mathrm{tw})}$ time whether $S$ is shattered by the vertices of $Y$. Recall that $|V(T)| = \mathcal{O}(|V|)$, thus, this phase takes $2^{\mathcal{O}(\mathrm{tw})} \cdot |V|$ time. If $|S| \geq k$ for any such shattered set $S$, then we stop and return YES. Otherwise, we move to the second phase.

**Second Phase.** We may now assume that any shattered set of size $k$ in $G$ is not entirely contained within any bag of $T$. By Lemma 15, if $\log \mathrm{tw} + 2 < k$, then we return NO. Otherwise, $k \leq \log \mathrm{tw} + 2$. We now show how we compute a shattered set of maximum size via dynamic programming on the tree-decomposition in $2^{\mathcal{O}(\mathrm{tw} \cdot \log \mathrm{tw})} \cdot |V|$ time, which dominates the runtime of the overall algorithm.

**Equivalent Formulation.** Let $S \subseteq X$ be a shattered set of $G$ such that $|S| = k$. Then, there exists a set $W \subseteq Y$ (of witnesses) such that $|W| = 2^k$ and, for each subset $A \subseteq S$, there is a unique vertex $w$ in $W$ that witnesses $A$, i.e., $N_S(w) = A$. Hence, finding a shattered set of size $k$ in $G$ is equivalent to finding $S \subseteq X$ and $W \subseteq Y$ such that $|S| = k$, $|W| = 2^k$, and $W$ witnesses $S$ (see Figure 2). Let us fix a labeling $\{s_1, \ldots, s_k\}$ of vertices in $S$ and a labeling $\{w_1, \ldots, w_{2^k}\}$ of vertices in $W$. This gives us a *pattern graph* $\mathcal{P}$ (the idea of a pattern graph was introduced in [30]). Now, if we can find a (induced) subgraph $H$ of $G$ with $|V(H)| \leq 2^k + k$ and a function $h : V(H) \to V(\mathcal{P})$ such that (i) for any $u, v \in V(H)$ satisfying $h(u) \in S$ and $h(v) \in W$, or vice versa, $uv \in E(H)$ if and only if $h(u)h(v) \in E(\mathcal{P})$, and (ii) for distinct $u, v \in V(H)$ satisfying $h(u), h(v) \in S$ or $h(u), h(v) \in W$, $h(u) \neq h(v)$; then it is easy to see that the vertices of $H$ mapped to $S$ via $h$ form a shattered set. Thus, in our dynamic programming algorithm, we look for a such a pattern and function in $G$.

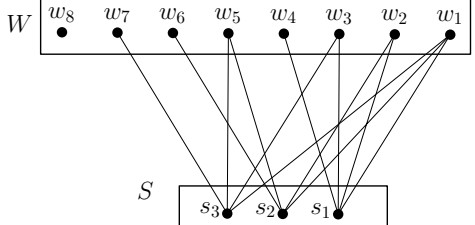

Figure 2: An illustration for the pattern graph $\mathcal{P}$. Here, $S \subseteq X$, $W \subseteq Y$, and $W$ witnesses $S$.

**Mapping Bags to the Pattern.** Consider a fixed labeling of $S \cup W = \{s_1, \ldots, s_k, w_1, \ldots, w_{2^k}\}$, and its corresponding pattern $\mathcal{P}$. Our algorithm will check if there is a subgraph of $G$ corresponding to this labeling and pattern. To this end, we define a dynamic programming table (DP table), for which the following function will be useful. For a node $t \in V(T)$, we have a function $f : V(\mathcal{P}) \to \beta(t) \cup \{\uparrow, \downarrow\}$, which assigns each vertex $x \in W \cup S$ to a vertex in the bag $\beta(t)$, or $\downarrow$, which means that the vertex $x$ is mapped to a vertex in $G_t^{\downarrow}$, or to $\uparrow$, which means that $x$ will be mapped to a vertex in $G_t^{\uparrow}$.

**Definition 16** (Valid function). *Let $t$ be a node of $T$. A function $f : V(\mathcal{P}) \to \beta(t) \cup \{\uparrow, \downarrow\}$ is valid if the following conditions hold for all $i \in [k]$ and $j \in [2^k]$:*

*1. If $f(s_i) \in \beta(t)$, then $f(s_i) \in X$. Similarly, if $f(w_j) \in \beta(t)$, then $f(w_j) \in Y$.*

*2. If $f(s_i), f(w_j) \in \beta(t)$, (i.e., $f(s_i) \notin \{\uparrow, \downarrow\}$ and $f(w_j) \notin \{\uparrow, \downarrow\}$), then $f(s_i)f(w_j) \in E(G)$ if and only if $s_i w_j \in E(\mathcal{P})$.*

*3. If $s_i w_j \in E(\mathcal{P})$, then neither $f(s_i) = \uparrow$ and $f(w_j) = \downarrow$ nor $f(s_i) = \uparrow$ and $f(w_j) = \downarrow$.*

*4. For distinct $i, i' \in [k]$ if $f(s_i), f(s_{i'}) \in \beta(t)$, then $f(s_i) \neq f(s_{i'})$. Similarly, for distinct $j, j' \in [2^k]$, if $f(w_j), f(w_{j'}) \in \beta(t)$, then $f(w_j) \neq f(w_{j'})$.*

**Observation 17.** *For each node $t$, there are at most $(\mathrm{tw}+2)^{k+2^k}$ possible functions, i.e., $2^{\mathcal{O}(\mathrm{tw} \cdot \log \mathrm{tw})}$ many functions of the form $f : V(\mathcal{P}) \to \beta(t) \cup \{\uparrow, \downarrow\}$.*

For our bottom-up dynamic programming on $(T, \beta)$, we use the notion of *extending valid functions*, which, intuitively, "lifts" a valid function $f : V(\mathcal{P}) \to \beta(t)$ to a partial solution of $G_t$ (if it exists).

**Definition 18** (Extending valid functions). *Let $t$ be a node of $T$ and $f : V(\mathcal{P}) \to \beta(t) \cup \{\uparrow, \downarrow\}$ a valid function. A function $g : V(\mathcal{P}) \to V(G_t) \cup \{\uparrow\}$ extends $f$ if for all $i \in [k]$ and $j \in [2^k]$:*

*1. If $f(s_i) \in \beta(t) \cup \{\uparrow\}$, then $g(s_i) = f(s_i)$, and if $f(w_j) \in \beta(t) \cup \{\uparrow\}$, then $g(w_j) = f(w_j)$. If $f(s_i) = \downarrow$, then $g(s_i) \in G_t^{\downarrow}$, and if $f(w_j) = \downarrow$, then $g(w_j) \in G_t^{\downarrow}$.*

*2. If $g(s_i), g(w_j) \in V(G_t)$, then $g(s_i)g(w_j) \in E(G)$ if and only if $s_i t_j \in E(\mathcal{P})$.*

*3. If $x, y \in V(\mathcal{P})$ are two distinct vertices such that $g(x), g(y) \in V(G_t)$, then $g(x) \neq g(y)$.*

**DP States.** To formally define our DP table, we define the *DP states*. Let $t$ be a node of $T$ and $f : V(\mathcal{P}) \to \beta(t) \cup \{\uparrow, \downarrow\}$ a function. Then, a *DP state* $\Gamma(t, f) \in \{0, 1\}$, where $\Gamma(t, f) = 1$ implies that $f$ is a valid function and there is a mapping $g : V(\mathcal{P}) \to V(G_t)$ that extends $f$, and $\Gamma(t, f) = 0$ implies that either $f$ is invalid or $f$ is valid but there is no mapping $g : V(\mathcal{P}) \to V(G_t)$ that extends $f$. Next, we explain how we compute our DP States in a bottom-up manner for our tree-decomposition.

**Leaf Node.** For each leaf node $t$, $\beta(t) = \emptyset$. Thus, the only valid function for $t$ is $f : V(\mathcal{P}) \to \{\uparrow\}$. Furthermore, $\Gamma(t, f) = 1$.

**Introduce Node.** Let $t$ be an introduce node and $c$ its unique child such that $\beta(t) = \beta(c) \cup \{v\}$ for some $v \in V(G)$. Two DP states $\Gamma(t, f)$ and $\Gamma(c, f')$ are *introduce compatible* if the following hold:

1. $f$ is a valid function for $t$, and $f'$ is a valid function for $c$.

2. If $f(x) = v$ (for some $x \in V(\mathcal{P})$), then $f'(x) = \uparrow$ and, for each $y \in V(\mathcal{P}) \setminus \{x\}$, $f(y) = f'(y)$.

We compute $\Gamma(t, f)$ as follows: $\Gamma(t, f) = \bigvee\limits_{f' \text{ is introduce compatible with} f} \{\Gamma(c, f')\}$.

**Forget Node.** Let $t$ be a forget node which has a unique child $c$ such that $\beta(c) = \beta(t) \cup \{v\}$ for some $v \in V(G)$. Two DP states $\Gamma(t, f)$ and $\Gamma(c, f')$ are *forget compatible* if the following hold:

1. $f$ is a valid function for $t$, and $f'$ is a valid function for $c$.

2. If $f'(x) = v$ (for some $x \in V(\mathcal{P})$), then $f(x) = \downarrow$ and, for each $y \in V(\mathcal{P}) \setminus \{x\}$, $f(y) = f'(y)$.

We compute $\Gamma(t, f)$ as follows: $\Gamma(t, f) = \bigvee\limits_{f' \text{ is forget compatible with} f} \{\Gamma(c, f')\}$.

**Join Node.** Let $t$ have exactly two children $c_1, c_2$, and let $\beta(t) = \beta(c_1) = \beta(c_2)$. The DP states $\Gamma(c_1, f_1)$ and $\Gamma(c_2, f_2)$ are *join compatible* with $\Gamma(t, f)$ if the following hold:

1. $f, f_1$, and $f_2$ are valid functions for $t, t_1$, and $t_2$, respectively.

2. For all $x \in V(\mathcal{P})$, if $f(x) \in \beta(t) \cup \{\uparrow\}$, then $f(x) = f_1(x) = f_2(x)$, and if $f(x) = \downarrow$, then either $f_1(x) = \downarrow$ and $f_2(x) = \uparrow$, or $f_1(x) = \uparrow$ and $f_2(x) = \downarrow$.

We compute $\Gamma(t, f)$ as follows: $\Gamma(t, f) = \bigvee\limits_{f_1, f_2 \text{ are join compatible with} f} \{(\Gamma(c_1, f_1) \wedge \Gamma(c_2, f_2))\}$.

For the root node $r$ of $T$, $\beta(r) = \emptyset$ and $G_r = G$. Also, note that there is a unique valid function $f : V(\mathcal{P}) \to \beta(r)$, i.e., for each $x \in V(\mathcal{P})$, $f(x) = \downarrow$. Now, $\Gamma(r, f) = 1$ implies that there is a function $g : V(\mathcal{P}) \to V$ that extends $f$, and the subset of vertices $S \subseteq V$ mapped to vertices in $\{s_1, \ldots, s_k\}$ is a shattered set and the subset of vertices $W \subseteq V$ mapped to vertices in $\{w_1, \ldots, w_{2^k}\}$ witnesses $S$. Similarly, if $\Gamma(r, f) = 0$, then there is no shattered set of size $k$ in $G$. Since (i) we consider at most $2^{\mathcal{O}(\text{tw} \cdot \log \text{tw})}$ possible functions for each node $t \in V(T)$ (Observation 17), (ii) all our operations over a function require polynomial time in the size of the (at most three) considered bags, and (iii) there are $\mathcal{O}(|V|)$ bags, we have that the running time of our second phase is $2^{\mathcal{O}(\text{tw} \cdot \log \text{tw})} \cdot |V|$. Since phase one requires $2^{\mathcal{O}(\text{tw})} \cdot |V|$ time, we have the following result.

**Theorem 19.** GEN-VC-DIMENSION *admits an algorithm running in* $2^{\mathcal{O}(\text{tw} \cdot \log \text{tw})} \cdot |V|$ *time.*

## 6 Conclusion

Computing the VC-dimension of a set system or graph has numerous applications in machine learning and other areas. We have advanced the understanding of its parameterized complexity, providing algorithms and conditional lower bounds for several important parameters, including the maximum degree, treewidth (tw), and vertex cover number (vcn). The most challenging problem left open by our work is to close the gap between the running time of our $2^{\mathcal{O}(\text{tw} \cdot \log \text{tw})} \cdot |V|$-time algorithm and our $2^{o(\text{vcn}+k)} \cdot |V|^{\mathcal{O}(1)}$ ETH-based lower bound for GEN-VC-DIMENSION. It would also be interesting to know whether our 1-additive FPT approximation algorithm for VC-DIMENSION can be improved to an exact FPT algorithm. Future work could also include studying the setting in which the set system is defined by a circuit, which allows the input size to be dependent only on the size of the domain in some cases [70]. Notably, our lower bound from Theorem 9 also holds in this setting.

## Acknowledgments and Disclosure of Funding

This work was supported by the French government IDEX-ISITE initiative 16-IDEX-0001 (CAP 20-25), the International Research Center "Innovation Transportation and Production Systems" of the I-SITE CAP 20-25, the ANR project GRALMECO (ANR-21-CE48-0004), the CNRS IRL ReLaX, the EU Horizon Europe TaRDIS project (grant agreement 101093006), and the INSPIRE Faculty Fellowship by DST, Govt of India. We thank Maël Dumas for comments on a preliminary version of this manuscript.

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
