# OpenReview forum: "The Parameterized Complexity of Computing the VC-Dimension"
_NeurIPS.cc/2025/Conference — NeurIPS 2025 poster_

### Official Review · Reviewer_KBz3 · 2025-06-05

**Clarity:** 3
**Significance:** 3
**Originality:** 3
**Rating:** 5
**Confidence:** 4

**Summary:**

This paper presents a systematic parameterized-complexity study of computing the VC-dimension of set systems (hypergraphs). Building on earlier results that established W[1]-hardness when parameterized by the solution size or the hypergraph degeneracy, the authors obtain several new findings:

- **Maximum degree.** It is 1-additive approximation FPT algorithm when parameterized by the maximum degree of the hypergraph.
- **Maximum hyperedge size.** It is FPT when parameterized by the largest hyperedge size.
- **Treewidth.** It is FPT when parameterized by the treewidth of the corresponding bipartite incidence graph.
- **Hypertree width / transversal number.** It is *not* FPT when parameterized by hypertree width or transversal number, unless LogNP=P

The paper also explores several variants of the VC-dimension problems.

**Questions:**

See "weaknesses"

**Ethical Concerns:**

["NO or VERY MINOR ethics concerns only"]

**Final Justification:**

This paper provides a comprehensive parameterized-complexity analysis of computing the VC-dimension of set systems. The results are technically sound, and the topic is highly relevant to the conference. I am positive about the work.

**Limitations:**

yes

**Paper Formatting Concerns:**

No major formatting issues

**Quality:**

3

**Strengths And Weaknesses:**

**Strengths:** By combining new algorithms with matching hardness for a wide range of structural parameters, this paper provides a comprehensive picture on the parameterized complexity of VC-dimension. The results are clearly presented and well organized.

**Weaknesses & Suggestions:** Several questions remain to be explored. For examples,  do the lower bounds still hold when one aims to approximate, rather than exact compute, the VC-dimension?  Clarifying this approximability landscape would further strengthen the contribution.

---

> ### Author Rebuttal · Authors · 2025-07-30
>
> Thank you for your helpful insights to improve the paper, and we are happy to hear that you appreciated the clarity and organization of our paper.
>
> We will elaborate the discussions around approximability. We recall that we already mentioned inapproximability results from the literature in lines 47-49 and 65-66. We would also get the existing inapproximability results from the literature in Proposition 13. For Theorem 9, we believe the reduction would need to be significantly changed in order to get inapproximability. As mentioned in our response to reviewer HyJm, an existing hardness result from the literature also provides inapproximability, showing that our FPT approximation algorithm is in some sense tight with respect to the choice of parameter. We will make the appropriate changes.

---

> > ### Comment · Reviewer_KBz3 · 2025-08-07
> >
> > Thanks for the response. I maintain my positive score.

---

> > > ### Author Response · Authors · 2025-08-07
> > >
> > > Thank you very much!

---

### Official Review · Reviewer_gAYP · 2025-06-23

**Clarity:** 3
**Significance:** 3
**Originality:** 3
**Rating:** 5
**Confidence:** 4

**Summary:**

The paper is a theoretical investigation of the complexity of computing the VC dimension. VC dimension is a key quantity with far-reaching implications so understanding the complexity is a fundamental task and with a long history.

The authors establish several results:

-Under ETH, the simple algorithm that runs in 2^n time is optimal. This algorithm, given a hypergraph on V vertices, will simply check all subsets of V to see if it's shattered. The main technique here is a clever reduction from the 3-coloring problem and relying on known hardness for the latter.

-Given this, the authors turn to fixed parameter approaches, and show that VC computation depends on the types of parameters used. For some natural choices of them, like maximum degree ∆ of hyperedges, they almost solve the problem by providing a additive-1 approximation. They also analyze the dimension D of the hypergraph, showing FPT algorithms, and also on the negative, other core structural hypergraph parameters (hypertree, width and transversal number) do not yield FPT algorithms.

-Finally, the authors focus on treewidth parameterization and show positive results. The define a slight generalization called GEN-VC-DIMENSION and show it is FPT parameterized by the treewidth tw of G (Theorem 17), in particular they design a 2^O(tw·log tw) · |V |^O(1)-time algorithm. They also show a lower bound based on vertex cover number vc which is always at least as large as tw.

**Questions:**

minor question that would improve the presentation and highlight novelties in your reductions: compared to the previous works doing FPT algos for VC dimension, it would be nice to have a paragraph or two where you compare not just the results from previous works to yours, but also the techniques as well. For example, the 3-coloring reduction, were there known connections between the two problems?

**Ethical Concerns:**

["NO or VERY MINOR ethics concerns only"]

**Final Justification:**

I have read the authors response and maintain my score.

**Limitations:**

-it is a theoretical work so yes.

**Paper Formatting Concerns:**

looks good.

**Quality:**

4

**Strengths And Weaknesses:**

STRENGTHS:
-they authors solve fundamental problems in the area that should be of broad interest
-deep set of results combining FPT knowledge and connecting it to VC dimension
-clever reduction from 3-Coloring to VC dimension computation, this leads to Theorem 9. This is satisfying result given that the naive algorithm appears to be best under ETH in some sense
-clever usage of treewidth in an algorithmic way to determine the VC dimension

WEAKNESSES:
-literature comparison at a technical level would be appreciated, see brief question below.

---

> ### Author Rebuttal · Authors · 2025-07-30
>
> Thank you for your kind review and nice suggestion to improve our paper. We are glad that you appreciated our results.
>
> We will elaborate on the discussion of the related work to add which techniques were used. We remark that there are very few parameterized algorithms for VC-dimension and just a few hardness results, and that was actually a main motivation for the paper. For example, we will highlight that the previous W[1]-hardness results came from reductions from CLIQUE and the previous FPT inapproximability came from a reduction from MAXIMUM BICLIQUE. As far as we know, there were no known prior connections between 3-COLORING and our problem.

---

### Official Review · Reviewer_42Hd · 2025-07-01

**Clarity:** 4
**Significance:** 3
**Originality:** 3
**Rating:** 5
**Confidence:** 3

**Summary:**

The authors introduce the VC dimension for graphs and hypergraphs and show a lower bound on the complexity of computing the VC dimension for graphs conditioned on the exponential time hypothesis.
They show this via an exponential time reduction from the 3-Coloring problem.
They further show results in parameterized complexity, for the parameters maximum degree and treewidth.
For the treewidth result, they provide an explicit dynamic programming approach.

**Questions:**

Could you please elaborate further on the exponential time reduction? Why can't we just solve 3Col in the reduction? Please elaborate why the use of an exponential time reduction is not problematic.

**Ethical Concerns:**

["NO or VERY MINOR ethics concerns only"]

**Final Justification:**

This is a solid paper. The authors have resolved my concerns in their rebuttals.

**Limitations:**

Yes.

**Paper Formatting Concerns:**

None.

**Quality:**

4

**Strengths And Weaknesses:**

+:
The paper provides a very nice introduction to the subject, such that even those outside the field can follow.
The proofs are written clearly, in a step-by-step manner, that is reasonably easy to follow, even considering their at times quite technical nature.

-:
While much of the paper is very clearly communicated, the reasoning and exact nature for the kind of exponential time reduction they use is not explicitly provided.
I believe that it would be very helpful to the reader to devote a short paragraph to explaining, why exponential time is needed, what kind of exponential time you are considering and why such a reduction gives you the result in the end.
Further, a longer conclusion, considering possible impact and tying your work to other results in the area could benefit the paper, although I do realize that space is limited.

minor comments:
l. 126: Maybe don't use T for the tree, since you used T as a symbol for a subset of V in the sentence before. Using another symbol for either of the Ts could help avoid confusion.
l. 199: it should be "i \in [k]" instead of "i \in [j]", right?
l. 294: "thesesubsets" -> "these subsets"
l. 300f: I don't understand the sentence inside the parentheses.
l. 369: "when it exists" -> "if it exists"

---

> ### Author Rebuttal · Authors · 2025-07-30
>
> Thank you for your nice review and useful suggestions. We are glad that you appreciated our presentation.
>
> Concerning why the reduction should require exponential time, we will expand on what was written in footnote 4 which appeared at the beginning of the reduction. In particular, we will make a new paragraph before the description of the reduction that explains why it is unlikely to have such a reduction take polynomial time. The reason is that such a polynomial-time reduction would imply that NP=QP due to 3-COLORING being NP-hard and VC-DIMENSION being in QP (solvable in quasi-polynomial time $|H|^{O(\log |H|)}$). This would be a monumental result as it is not known whether NP=QP and it is widely believed that they are not equal.
>
> In the reduction description, we will also explicitly state the exponential time the reduction takes (we note that it was already mentioned in the proof of Lemma 8). Lastly, to further clarify why such a reduction gives us the desired result, we will expand on the first sentence of the paragraph before Theorem 9.
>
> Also, an exponential-time reduction is not problematic in this case since the aim is to exclude the possibility of a certain type of exponential runtime. Thus, all that we require is that the time the reduction takes is at most the exponential runtime that we want to exclude. This is standard for ETH-based reductions just as polynomial time is for NP-hardness reductions, and FPT time is for parameterized reductions (W[1]-hardness).
>
> We cannot just solve 3-COL in the reduction since by Lemma 8, solving it in the running time that we are trying to exclude for VC-DIMENSION would contradict Proposition 1.
>
> Concerning the conclusion, we will try to extend it as much as possible. As mentioned by other reviewers, it should be a good place to discuss more future directions, such as the case where the set system is defined implicitly rather than explicitly.
>
> Lastly, thank you for pointing out the typos and unclear sentence in the minor comments. We will make the appropriate changes.

---

> > ### Comment · Reviewer_42Hd · 2025-08-01
> > **Response to the rebuttal**
> >
> > Thank you for the clarification. I am satisfied with the promised revision and will maintain my rating.

---

> > > ### Author Response · Authors · 2025-08-02
> > >
> > > Thank you very much!

---

### Official Review · Reviewer_HyJm · 2025-07-03

**Clarity:** 3
**Significance:** 2
**Originality:** 3
**Rating:** 4
**Confidence:** 4

**Summary:**

This paper studies the problem of computing the VC dimension of a set system. Let us recall the set up. A set system can be represented as a hypergraph or the corresponding bipartite incidence graph. A subset S of the universe is called "shattered" if, for every subset T of S, there is a set in the system whose intersection with S is exactly T. So, in some sense, every subset of S "appears" if we restrict the universe (of the set system) to be S.

The VC dimension is the size of largest shattered set. The computational problem is to compute the VC dimension given an explicit representation of the set system. Note that the largest possible shattered set has size log H, where H is the number of sets in the system. So that automatically gives H^(log H) time algorithms. This is essentially equivalent to also trying out every subset of the system and checking it is shattered.

There are a series of results:

Result 1. Assuming the Exponential Time Hypothesis, all exact algorithms require 2^{(\Omega(|V|)} time, where V is the universe.
Result 2. There is a Delta^D poly(|H|) time algorithm that gives an additive 1-approximation to the VC dimension. Delta is the max degree of the incidence graph.
Result 3. There is a (tw)^(tw) poly(H) algorithm, where tw is the treewidth of the incidence graph.

Result 1 comes from a reduction from 3-coloring. It goes via a graph VC dimension problem, where the set system is given by the neighborhoods.
Result 2 comes from ordering ideas (reminiscent of graph degeneracy).
Result 3 is the most technical, and essentially constructs a dynamic programming solution using the tree decomposition (the typical paradigm).

**Questions:**

Major:

1. Are there any use cases/applications where one wants to compute the VC dimension of an explicit set system? I think the paper would be stronger with a more compelling case for this problem. I consider NeurIPS a non (purely) theoretical venue. So a purely theoretical justification (as a TCS paper) doesn't seem enough.
2. Can one improve the Delta^D poly(|H|) to get a dependence on the degeneracy? Certainly seems possible.

Minor:

1. I contest the point that this paper shows that the trivial 2^{(O(|V|)} algorithm is optimal. I suggest that be reworded.
2. Prop 1 "Unless the ETH fails, there exists an.." has double negatives and is hard to parse. I suggest: "Assume the ETH. Then there exists a constant \eps such that all algorithms have running time \Omega(...).

**Ethical Concerns:**

["NO or VERY MINOR ethics concerns only"]

**Limitations:**

Yes

**Quality:**

3

**Strengths And Weaknesses:**

The VC dimension is a fundamental quantity of interest, and it's nice to have a suite of results. The first result on the exponential hardness sets the stage for parameterized algorithms. And both the parameters (degree and treewidth) are natural enough in this setting.

My primary complaints about the paper are as follows:
 - (Major) Is this the right problem/setting to study? In most VC dimension settings, the number of sets is extremely large and implicitly defined (think halfspaces). So there is often a circuit that describes the set systems. The algorithms described do not (or do they?) port to that setting. It is very rare that one has an explicit set system and wishes to compute the VC dimension.
 - (Minor) It's incorrect to say that the 2^O(|V|) algorithm has been shown to be optimal, assuming ETH. The hardness does not rule out exponential time algorithms with a smaller exponent, which would certainly be useful. The hardness shows that it is unlikely, we can get subexponential dependence.

---

> ### Author Rebuttal · Authors · 2025-07-30
>
> Thank for your positive review and helpful comments to improve the paper.
>
> Major Questions:
>
> 1. We think that you bring up an interesting question. It is true that when the set system is very large, it is more efficient to represent it as a circuit. However, as pointed out in [1], it is also true that for many set systems, representing them as a circuit can only be done through an exponential blowup in the circuit, negating its utility.
>
> Our work is clearly a theoretical work (as with many other past NeurIPS papers) and we do not claim any immediate real-world applications of our algorithms. As can be seen in our introduction, there are a large number of works that consider the same explicit representation setting as us. Thus, of course there is a choice of which setting to study, but we do not feel that there are very strong arguments to choose one over the other: both are important. It is also difficult to study both in a single work due to space limitations, which is exemplified by past papers on the topic.
>
> Nevertheless, we believe that our algorithms may still be useful in some practical settings, especially when the set system (whether provided explicitly or implicitly) has size polynomial in the universe $\mathcal{V}$. For example, this is the case for set systems of bounded VC-dimension (due to the Sauer-Shelah lemma) or when the set system is a graph. See [2] for a recent work computing the VC-dimension of graph neural networks (GNNs) where they upper bound the VC-dimension by functions of other parameters of the GNN such as the bitlength of its weights. See also [3,4] for recent experimental works computing the VC-dimension of graphs.
>
> Moreover, we do want to point out that our main lower bound result (Theorem 9) also applies to the implicit setting, providing a very valuable lower bound to both settings. Indeed, the excluded runtime is $2^{o(|\mathcal{V}|)}\cdot poly(|\mathcal{V}|)$ since $|H|=poly(|\mathcal{V}|)$ in our reduction.
>
> Concerning our algorithms, they do not make much sense in the implicit representation of the set system setting since a tree decomposition (whose size depends on the size of the set system) needs to be constructed for our treewidth-based algorithm and we also use the fact that it can be checked whether a set can be shattered in polynomial time in the explicit setting, which is very unlikely to be true in the implicit setting since computing the VC-dimension when the set system is represented by a circuit is $\Sigma^p_3$-complete (i.e., most likely not in NP) [5]. We believe the latter fact is a big stumbling block that probably rules out most FPT algorithms in the circuit setting, possibly implying that the circuit setting is simply too intractable to be of significant interest from a parameterized complexity perspective. We think it would be an interesting direction to validate or invalidate this intuition in future work, and perhaps there are useful parameterizations that arise from the circuit itself that could be investigated.
>
> 2. This is an interesting question, which did not occur to us. Our main aim was to get as close as possible to an FPT algorithm parameterized by the maximum degree since it is known that the problem is W[1]-hard parameterized by the degeneracy (as mentioned in our paper, see reference [29]). However, after checking the literature, it seems that it is not possible to have such an FPT approximation algorithm parameterized by the degeneracy as the aforementioned W[1]-hardness reduction should also rule this out as it is from CLIQUE which cannot be o(OPT)-FPT approximated under the Gap-ETH [5].
>
> We will incorporate these remarks, which nicely improve the paper.
>
> Minor Questions:
>
> 1. Concerning the remark about the $2^{O(|\mathcal{V}|)}$ algorithm being optimal, assuming the ETH, we will change "optimal" to "asymptotically tight" to avoid any confusion since of course the big O and little o notations allow for different constants.
>
> 2. We will change the statement of Proposition 1 accordingly.
>
> References:
>
> [1] M. Schaefer. Deciding the Vapnik-Cervonenkis Dimension is $\Sigma^p_3$-Complete. J. Comput. Syst. Sci. 58(1): 177-182, 1999.
>
> [2] C. Morris et al. WL meet VC. 40th International Conference on Machine Learning (ICML 2023), pp. 25275-25302, 2023.
>
> [3] D. Coudert et al. Practical Computation of Graph VC-Dimension. 22nd International Symposium on Experimental Algorithms (SEA 2024): 8:1-8:20, 2024.
>
> [4] P. Drange et al. Computing complexity measures of degenerate graphs. 18th International Symposium on Parameterized and Exact Computation (IPEC 2023): 14:1–14:21, 2023.
>
> [5] P. Chalermsook et al. From Gap-ETH to FPT-Inapproximability: Clique, Dominating Set, and More. 2017 IEEE 58th Annual Symposium on Foundations of Computer Science (FOCS 2017), pp. 743-754, 2017.

---

### Decision · Program_Chairs · 2025-09-17

**Decision:**

Accept (poster)

**Comment:**

This is a well-written paper that studies the parameterized complexity of computing the VC-dimension, a fundamental notion in learning theory and combinatorics. The paper provides both algorithmic and hardness results under natural structural parameters such as maximum degree, hyperedge size, and treewidth.

We recommend acceptance as a poster. The work is technically solid and addresses a fundamental problem with clarity and rigor.

That said, we would like to point out that some doubts were raised during the discussion regarding whether NeurIPS is the most appropriate venue for this type of result. While the VC-dimension plays a central role in understanding generalization in machine learning, the algorithmic problem of computing the VC-dimension is less common in practical ML applications.

We thank the authors for their contribution.